# Comparing Insomnia and Perceived Stress in Online vs. Frontal Learning: Psychometric Evaluation in a Health Sciences Cohort

**DOI:** 10.3390/healthcare13182272

**Published:** 2025-09-11

**Authors:** Johanna Andrea Márton, Flóra Busa, Nóra Rozmann, Attila András Pandur, Melinda Petőné Csima, David Sipos

**Affiliations:** 1Doctoral School of Health Sciences, Faculty of Health Science, University of Pécs, Vörösmarty Str. 4, 7621 Pẻcs, Hungary; 2Department of Medical Imaging, Faculty of Health Sciences, University of Pécs, Vörösmarty Str. 4, 7621 Pẻcs, Hungary; 3Department of Oxyology, Emergency Care, Faculty of Health Sciences, University of Pécs, Vörösmarty Str. 4, 7621 Pécs, Hungary; 4Institute of Education, MATE-Hungarian University of Agriculture and Life Sciences, Guba Sándor Street 40, 7400 Kaposvár, Hungary

**Keywords:** mental health, student health, sleep quality, perceived stress, health science students, online education, frontal learning

## Abstract

**Background/Objectives**: University students in health sciences are particularly vulnerable to poor sleep quality and elevated stress due to academic and clinical demands. This follow-up study aimed to compare sleep quality and perceived stress levels among health science students during online and frontal (in-person) education periods and to examine the influence of behavioral and demographic factors. **Methods**: A prospective, follow-up design was applied involving students from the University of Pécs, Faculty of Health Sciences, across nursing, radiography, laboratory analytics, paramedicine, and physiotherapy programs. Data were collected via anonymous online questionnaires during two intervals: February–March 2023 (online education) and April–May 2023 (frontal education). The Athens Insomnia Scale (AIS) and Perceived Stress Scale (PSS-10) were used to assess sleep quality and stress, respectively. Internal consistency was evaluated using Cronbach’s alpha (AIS: α = 0.81–0.84; PSS: α = 0.87–0.90). Data were analyzed using descriptive statistics and paired sampled *t*-tests (*p* < 0.05). **Results**: AIS scores were significantly higher during online learning compared to in-person (5.47 ± 2.67 vs. 4.25 ± 2.48; *p* = 0.001), indicating poorer sleep quality. In contrast, PSS scores were higher during the frontal period (29.48 ± 8.67 vs. 24.31 ± 7.15; *p* < 0.05). Increased screen time, irregular routines, and lack of physical activity were associated with poorer outcomes. **Conclusions**: Online education may compromise sleep quality, while in-person learning appears to increase perceived stress. These findings highlight the need for targeted health promotion strategies adapted to different educational modalities.

## 1. Introduction

Sleep quality and stress are critical factors influencing the well-being and academic performance of university students, particularly those in health science disciplines. These students often face high academic workloads, clinical responsibilities, and emotionally taxing experiences, which contribute to elevated stress levels and disrupted sleep patterns. Poor sleep quality among this population is associated with decreased cognitive performance, increased anxiety, and reduced resilience to stress [1,2,3].

A growing body of evidence indicates a bidirectional relationship between stress and sleep quality. Stress impairs the ability to initiate and maintain sleep, while insufficient or poor-quality sleep exacerbates stress responses [4,5]. Health science students may be especially vulnerable to this cycle due to the demanding nature of their training and frequent exposure to high-pressure environments [6,7].

The ubiquitous use of smart devices adds a further dimension to this issue. Mobile phones, tablets, and laptops are frequently used by students for academic and recreational purposes, often during evening hours [8]. These devices emit blue light that disrupts circadian rhythms by suppressing melatonin production, leading to delayed sleep onset and reduced sleep efficiency [9]. Additionally, psychological stimulation from digital content and constant connectivity can contribute to increased stress levels and cognitive arousal before bedtime [8,9,10].

Educational delivery methods also influence students’ stress and sleep. Traditional in-person (frontal) instruction offers structured routines and face-to-face interaction but can increase time-related pressures due to commuting and fixed schedules [11]. Online learning provides flexibility but may lead to irregular sleep–wake cycles, increased screen time, and social isolation. Past research by Busa et al. [6] examined these factors in a cross-sectional context; however, the present follow-up study expands on that work by employing a longitudinal design to compare the same group of students across two distinct teaching modalities—online and traditional—thereby allowing for direct assessment of changes in stress and sleep patterns over time. The rationale for focusing on these two modes lies in their differing potential impacts; traditional teaching is expected to promote better sleep quality due to more regular schedules and reduced evening screen exposure, whereas online learning—particularly asynchronous formats—is anticipated to increase stress and impair sleep through prolonged screen time, irregular daily routines, and reduced social interaction. The shift between these modalities, especially in the context of hybrid education models, can disrupt students’ daily rhythms and contribute to both elevated stress and poor sleep hygiene [11,12,13].

Understanding the combined impact of academic stressors, smart device use, and instructional modalities on sleep quality is essential for informing educational policy and promoting student well-being [14,15]. Health science students represent a high-risk group in this context, making targeted research and intervention particularly necessary [1,2,3,6,9].

The primary aim of this study was to investigate longitudinal changes in sleep quality and perceived stress among health science students at the University of Pécs, Hungary, across two academic time points. Specifically, the study sought to examine how academic stressors, smart device usage, and variations in educational delivery methods (in-person vs. online learning) influence students’ sleep patterns and stress levels over time. By identifying the interplay between these factors, this study aims to provide evidence to inform educational practices and promote targeted interventions to enhance student well-being and academic performance.

## 2. Materials and Methods

This follow-up study was conducted at the Faculty of Health Sciences, University of Pécs, Hungary, targeting students enrolled in various health science programs, including radiography, laboratory analytics, nursing, paramedicine, and physiotherapy. The aim was to assess changes in sleep quality and perceived stress over time.

### 2.1. Study Design and Participants

Data collection occurred in two phases using an anonymous, self-administered online questionnaire hosted on Google Forms. The study period coincided with two distinct teaching modalities: (1) a fully online teaching period in 1 February–1 March 2023 and (2) a traditional, in-person teaching period in 1 April–26 May 2023. The online teaching phase was primarily asynchronous, with pre-recorded lectures and uploaded materials complemented by occasional synchronous sessions for discussion and clarification, lasting approximately 60–90 min each. The traditional teaching phase consisted of in-person lectures, seminars, and practical classes scheduled according to the university timetable, with an average daily duration of 4–6 h. Students were therefore exposed to the online teaching modality for approximately 4 weeks before transitioning to traditional teaching.

The adequate sample size was calculated using a priori power analysis (G*Power version 3.1), assuming a medium effect size (Cohen’s d = 0.5), an alpha level of 0.05, and a statistical power of 0.80 for paired-sample comparisons. This yielded a minimum required sample size of 34 participants.

There were 2012 active students in the Faculty of Health Sciences at the University of Pécs during the survey. A total of 304 students met the inclusion criteria (being actively enrolled in the targeted programs and willing to participate in both phases). Of these 97 students (31.9%) completed both phases.

The first data collection period was conducted between February and March 2023, followed by a second wave between April and May 2023. Participation was voluntary, and informed consent was obtained from all respondents prior to completing the questionnaire. Inclusion criteria consisted of being an actively enrolled student in the aforementioned programs and consenting to participate in both phases of the study.

### 2.2. Instruments

Two validated tools were used to assess sleep quality and perceived stress.

#### 2.2.1. Athens Insomnia Scale (AIS)

The Athens Insomnia Scale (AIS) is a psychometrically validated, self-administered instrument consisting of eight items specifically developed to evaluate the severity of insomnia symptoms in accordance with the diagnostic criteria outlined in the International Classification of Diseases, 10th Revision (ICD-10). Each item is rated on a four-point Likert scale ranging from 0 (no problem) to 3 (very serious problem), reflecting the intensity or frequency of specific sleep-related difficulties [16].

The first five items assess nocturnal sleep disturbances, including sleep induction, awakenings during the night, final awakening, total sleep duration, and overall sleep quality. The remaining three items evaluate the impact of poor sleep on daytime functioning, such as well-being, functioning capacity, and sleepiness during the day [16].

The total AIS score ranges from 0 to 24, with higher scores indicating greater severity of insomnia. A cumulative score of 6 or more is commonly employed as the diagnostic threshold for identifying clinically significant insomnia. This cut-off point has been supported by empirical research demonstrating its sensitivity and specificity in both clinical and general populations [16].

#### 2.2.2. Perceived Stress Scale (PSS)

The Perceived Stress Scale (PSS) is a widely used psychological instrument designed to measure the degree to which individuals perceive situations in their lives as stressful, particularly over the past month. The 10-item version of the scale (PSS-10) evaluates the extent to which respondents appraise their lives as unpredictable, uncontrollable, and overloaded—core components of the stress experience [17].

Each item is rated on a 5-point Likert scale ranging from 0 (“never”) to 4 (“very often”), reflecting the frequency with which the respondent has experienced certain thoughts and feelings related to stress. Items 4, 5, 7, and 8 are positively stated and are therefore reverse scored to reduce response bias and ensure that the total score accurately reflects perceived stress [17].

The total score ranges from 0 to 40, with higher scores indicating greater levels of perceived stress. The PSS-10 has demonstrated good psychometric properties across diverse populations and is considered a reliable and valid tool for assessing perceived stress in both research and clinical contexts [17].

#### 2.2.3. Self-Designed Questionnaire

In addition to standardized instruments such as the Athens Insomnia Scale (AIS) and the Perceived Stress Scale (PSS), this study employed a self-designed questionnaire to capture relevant lifestyle and behavioral factors potentially influencing sleep quality and perceived stress among participants. This custom questionnaire was developed to complement existing tools by gathering contextual data on variables not fully addressed by standard psychometric instruments.

The self-developed section of the questionnaire comprised categorical and multiple-choice items, each aimed at assessing daily habits and digital media use patterns. Specifically, it included the following domains:-Daily time spent watching TV or movies (categorized as max 1 h, 1–2 h, more than 2 h, and does not watch daily);-Duration of internet use before falling asleep (does not use, 10–15 min, more than 30 min, and more than 60 min);-Average daily screen time using a PC, laptop, or tablet (max 1 h, 1–2 h, and more than 2 h).

Additionally, socio-demographic variables were collected, such as gender, family status (single or in a relationship), academic performance (based on average grades), and frequency of sports activity (regularly, rarely, and does not do sports).

The self-designed questionnaire was administered both before and after the intervention period, allowing for the comparison of pre- and post-intervention data. This structure enabled the evaluation of potential behavioral influences on insomnia severity and perceived stress, as well as the identification of subgroups with differential outcomes.

Although not formally validated, the questionnaire was designed following best practices in survey development, with clearly defined response categories and face validity supported by relevant literature. Its integration with validated scales enhances the ecological validity of the study and provides richer insight into modifiable lifestyle factors associated with sleep and stress among the study population.

Prior to implementation, the questionnaire underwent expert review by three faculty members with expertise in sleep research, behavioral science, and survey design to ensure content validity. Items were then piloted on a convenience sample of 12 health science students to assess clarity, completion time, and technical functionality in the online format. Based on feedback, minor adjustments were made to item wording for clarity. Internal consistency reliability for the main behavioral domains was examined in the pilot sample using Cronbach’s alpha, which yielded acceptable values (α = 0.72–0.81).

### 2.3. Pairing the Students’ Responses

Prior to completing the questionnaire, students were instructed to generate a unique eight-character identification code. This code was used to anonymously match their responses across the two data collection periods. The code was constructed using the following elements: the first and last letters of the student’s first name, the first and last letters of the student’s mother’s first name, the last two digits of the student’s birth year, and the numerical day of their birth date. This method ensured the confidentiality of participants while allowing for longitudinal analysis. As noted above, unmatched cases due to incorrect or incomplete code entry were excluded from paired analysis but retained for phase-specific descriptive statistics.

### 2.4. Data Analysis

Data were analyzed using IBM SPSS Statistics for Windows, Version 23.0 (IBM Corp., Armonk, NY, USA). Descriptive statistics (mean, standard deviation, frequencies, and percentages) were used to summarize the data. For comparisons between groups and time points, paired-samples *t*-test was applied. A *p*-value of <0.05 was considered statistically significant.

### 2.5. Ethical Considerations

This study was approved by the Regional Research Ethics Committee (9633—PTE 2023). All data were collected and stored in accordance with the General Data Protection Regulation (GDPR) of the European Union.

## 3. Results

A total of 97 health science students participated in both phases of the follow-up study. The Athens Insomnia Scale (AIS) and Perceived Stress Scale (PSS) scores were compared across various demographic and behavioral variables between the initial (February–March 2023) and follow-up (April–May 2023) assessments.

Internal consistency was confirmed for both instruments at both time points. AIS: α = 0.81 (pre), 0.84 (post); PSS: α = 0.87 (pre), 0.90 (post), indicating good to excellent reliability across the follow-up period.

During the period of online education, participants reported significantly higher levels of insomnia symptoms, as measured by the Athens Insomnia Scale (AIS). The mean AIS score was 5.47 (SD = 2.67) during online instruction, compared to 4.25 (SD = 2.48) during the frontal (in-person) educational period (*p* = 0.001; Cohen’s d = 0.47, indicating a medium effect). In contrast, perceived stress levels, as assessed by the Perceived Stress Scale (PSS), were significantly higher during the frontal education period. The mean PSS score during in-person instruction was 29.48 (SD = 8.67), compared to 24.31 (SD = 7.15) during the online period (*p* < 0.05; Cohen’s d = 0.64, indicating a medium to large effect) (Figure 1).

In terms of gender differences, female students reported higher baseline AIS scores (M = 5.85, SD = 2.42) compared to males (M = 4.99, SD = 3.01). Post-intervention AIS scores decreased in both groups, more markedly among females (M = 4.16, SD = 2.21). PSS scores increased significantly for male students from pre-test (M = 20.47, SD = 7.97) to post-test (M = 28.91, SD = 7.12; *p* < 0.05).

Participants who reported no regular sports activity had significantly higher AIS scores pre-intervention (M = 6.23, SD = 3.47) and elevated PSS scores post-intervention (M = 33.11, SD = 10.25) compared to their peers engaging in regular physical activity (*p* < 0.05).

Regarding media consumption, students who spent more than two hours daily watching TV or movies had significantly higher pre-test AIS (M = 6.77, SD = 4.11) and post-test PSS scores (M = 37.27, SD = 9.90) compared to those with less screen time (*p* < 0.05).

Internet use before sleep also had a measurable impact. Participants using the internet for more than 60 min before falling asleep had significantly higher AIS scores pre-intervention (M = 6.99, SD = 4.11) and PSS scores pre-intervention (M = 38.49, SD = 9.79) compared to those who did not use the internet at bedtime (*p* < 0.05).

Excessive use of digital devices (PCs, laptops, and tablets) for more than two hours per day was associated with significantly elevated AIS (M = 6.24, SD = 3.71) and PSS scores (M = 35.12, SD = 9.74) at baseline (*p* < 0.05), suggesting a correlation between screen exposure and both insomnia symptoms and perceived stress (Table 1).

## 4. Discussion

This follow-up study explored the differential impact of educational modalities—online (remote) versus frontal (in-person) instruction—on sleep quality and perceived stress among health science students. Using validated psychometric instruments, including the Athens Insomnia Scale (AIS) and the Perceived Stress Scale (PSS), the investigation aimed to quantify the psychological and behavioral consequences of shifting learning environments in a post-pandemic academic context.

Our findings revealed that sleep quality was significantly poorer during online education, as evidenced by higher AIS scores (mean = 5.47, SD = 2.67) compared to the in-person period (mean = 4.25, SD = 2.48; *p* = 0.001). This deterioration aligns with prior research documenting the adverse effects of remote learning on students’ sleep hygiene. Multiple studies conducted during the COVID-19 pandemic highlighted that increased screen time, disrupted circadian rhythms, and reduced physical activity—all common features of online learning—contribute to the onset or exacerbation of insomnia symptoms [18,19]. Irregular daily routines, unstructured schedules, and the absence of fixed class times in asynchronous online settings can further delay sleep onset and impair sleep consolidation, as students may shift bedtimes later into the night. Excessive evening device use, especially for streaming lectures or completing assignments, not only exposes students to alerting blue light but also sustains cognitive arousal, making sleep initiation more difficult.

Alfonsi et al. emphasized that the transition to online modalities was associated with delayed bedtimes, extended sleep onset latency, and increased sleep dissatisfaction among university students [18]. Similarly, Marelli et al. reported that students experienced significant sleep–wake disturbances during remote instruction periods, with long-term consequences for mental and physical health [19].

Conversely, perceived stress levels were significantly higher during in-person education, with mean PSS scores rising from 24.31 (SD = 7.15) in the online period to 29.48 (SD = 8.67) during the frontal phase (*p* < 0.05). This suggests that, while the virtual environment may negatively affect sleep patterns, the structured and performance-oriented nature of in-person instruction imposes a higher psychological burden. Institutional demands such as mid-term and final examinations, continuous assessment requirements, mandatory clinical rotations, and practical skill evaluations often cluster during in-person semesters. These obligations, combined with commuting times and rigid timetables, can compress students’ available rest periods and elevate stress. Moreover, clinical placements in hospital or laboratory settings may introduce emotional stressors through patient care responsibilities, exposure to critical medical situations, and the pressure to perform competently under supervision.

As summarized in Table 2, these modality-specific differences highlight that, while online education primarily compromises sleep quality, in-person instruction predominantly elevates stress, each presenting unique risks and demands for students.

This observation is consistent with the prior literature demonstrating elevated stress levels among medical and health science students during face-to-face instruction. Dyrbye et al. found that such students are particularly vulnerable to burnout and stress due to high academic demands, competitive environments, and clinical obligations [20]. The need to meet academic and social expectations within rigid institutional schedules may contribute to heightened stress in the frontal modality [20,21].

Subgroup analyses further elucidated the role of behavioral and demographic variables in moderating stress and sleep outcomes. Male participants showed a significant post-instruction increase in PSS scores, and students who did not engage in regular physical activity or used digital devices for more than 60 min before bedtime experienced the highest levels of insomnia and stress.

These findings are corroborated by Exelmans and Van den Bulck, who demonstrated that prolonged screen exposure before sleep—particularly from smartphones or tablets—is linked to delayed sleep onset and decreased sleep efficiency [22]. Likewise, physical inactivity has been consistently associated with elevated stress and poor sleep outcomes across academic populations [23].

Interestingly, academic performance (i.e., grade average) did not significantly predict variations in AIS or PSS scores. This suggests that students’ perceived academic stress and behavioral coping mechanisms—rather than actual academic outcomes—may be more critical determinants of psychological distress. Ineffective time management and lifestyle imbalance, rather than GPA, were primary contributors to stress among university students [24,25,26].

The dual pattern observed—increased insomnia during online learning and elevated stress during in-person instruction—reflects the distinct psychosocial risks embedded within each educational modality. Online learning, though offering flexibility, may blur the boundaries between academic and personal time, foster excessive screen use, and disturb circadian regularity. In contrast, frontal instruction reintroduces institutional pressures, peer comparisons, and time constraints, which can amplify psychological strain.

Taken together, these findings underscore the need for context-sensitive mental health strategies that address the specific demands of each learning environment (Table 2). Institutional policies should prioritize sleep hygiene education, promote physical activity, and offer flexible academic supports to buffer stress in both online and in-person settings.

### 4.1. Limitations

Several limitations should be considered when interpreting the findings of this study. As the study relied on self-reported measures, the results may be influenced by recall bias, social desirability bias, and subjective misperceptions, which can lead participants to underreport undesirable behaviors (e.g., excessive screen use) or overreport socially valued ones (e.g., healthy sleep habits). Although the use of validated tools such as the AIS and PSS enhances reliability, the lack of objective sleep measures (e.g., actigraphy and sleep diaries) limits the precision of sleep-related outcomes.

Similarly, physiological stress markers (such as cortisol levels or heart rate variability) were not assessed, restricting the ability to validate perceived stress through biological indicators. This limits the depth of interpretation regarding physiological stress responses.

This study also did not account for several potential confounding variables, including socioeconomic status, employment, housing conditions, and pre-existing mental health or sleep disorders, all of which may have influenced the results. Their omission may affect the robustness of observed associations.

Moreover, the sample was confined to health science students, which may limit the generalizability of the findings to other student populations or non-academic groups. The specific academic and clinical demands faced by this population may not reflect broader student experiences.

Finally, the observational and non-randomized design of the study prevents causal conclusions. The sequential nature of online versus frontal learning phases may have introduced temporal or contextual confounds, such as seasonal variation or post-pandemic transitions, that were not controlled for.

### 4.2. Implications and Future Directions

Our findings underscore the need for educational institutions to adopt holistic strategies to protect student well-being across learning environments. For online education, this includes implementing sleep hygiene education, digital wellness programs, and promoting regular routines. For in-person learning, integrating stress-reduction interventions, such as mindfulness or peer support groups, may mitigate psychological strain. Future research should explore these interventions in longitudinal and controlled designs, incorporating both subjective and objective outcome measures.

## 5. Conclusions

This follow-up study highlights the differential effects of educational modality on sleep quality and perceived stress among health science university students. Online education was associated with significantly higher levels of insomnia, likely influenced by disrupted routines and increased screen exposure. In contrast, frontal (in-person) instruction was linked to elevated stress, possibly due to increased academic demands and social pressures. Additionally, lifestyle factors such as lack of physical activity and prolonged evening screen use were associated with poorer sleep and higher stress across both periods.

These findings underscore the need for targeted interventions tailored to the learning environment. Sleep hygiene education, digital behavior management, and structured stress-reduction programs should be considered essential components of university health promotion strategies. Practical implications for university health services include implementing workshops on healthy sleep habits, offering digital wellness programs that address evening screen use, and providing accessible stress management resources—particularly during peak academic and clinical periods.

As hybrid learning models become more prevalent, universities must proactively address the mental and physical well-being of students by integrating preventive and supportive measures within academic systems. It should be noted that these results are specific to a single institution and a health science student population and may not be generalizable to students from other disciplines or universities.

Further longitudinal studies involving objective measures and broader contextual variables are warranted to better understand the complex interplay between education, behavior, and student health.

## Figures and Tables

**Figure 1 healthcare-13-02272-f001:**
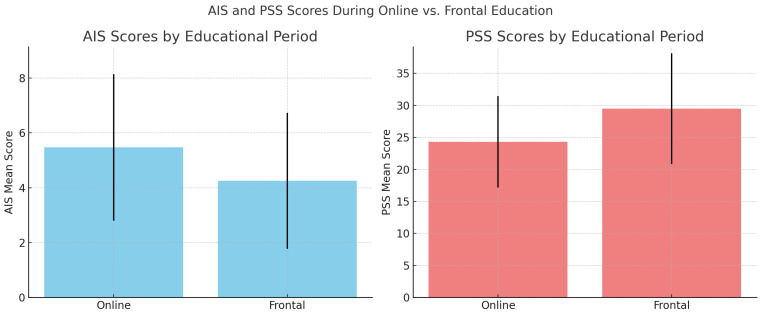
Comparison of AIS and PSS scores during online and frontal education periods.

**Table 1 healthcare-13-02272-t001:** AIS and PSS scores by demographic and behavioral variables. Mean (±SD) values measured before (PRE) and after (POST) follow-up by gender, family status, academic performance, and screen-related habits.

	n (%)	AIS PRE	AIS POST	Cohen’s d; *p* Value	PSS PRE	PSS POST	Cohen’s d; *p* Value
Gender							
Female	69 (71.1)	5.85 (SD = 2.42)	4.16 (SD = 2.21)	0.521; 0.142	28.24 (SD = 9.73)	29.67 (SD = 9.93)	−0.145; 0.548
Male	28 (28.9)	4.99 (SD = 3.01)	4.37 (SD = 2.71)	0.216; 0.425	20.47 (SD = 7.97)	28.91 (SD = 7.12) *	−1,117; 0.001
Family status							
Single	47 (48.4)	5.08 (SD = 3.11)	4.16 (SD = 2.57)	0.322; 0.124	24.61 (SD = 9.11)	29.51 (SD = 9.11)	−0.538; 0.247
In a relationship	50 (51.6)	6.01 (SD = 3.41)	5.20 (SD = 2.41)	0.274; 0.177	24.01 (SD = 10.14)	29.54 (SD = 9.57)	−0.561; 0.228
Average of the Previous Semester							
Medium	11 (11.3)	6.12 (SD = 3.49)	4.91 (SD = 2.20)	0.399; 0.343	24.72 (SD = 10.12)	29.98 (SD = 9.72)	−0.530; 0.221
Excellent	86 (88.7)	5.11 (SD = 3.12)	4.02 (SD = 2.28)	0.457; 0.214	24.06 (SD = 9.11)	28.96 (SD = 8.74)	−0.549; 0.078
Frequency of Sports							
Regularly	44 (54.3)	5.01 (SD = 3.11)	4.21 (SD = 2.77)	0.387; 0.097	20.49 (SD = 9.09)	25.49 (SD = 10.09)	−0.521; 0.086
Rarely	39 (40.5)	5.13 (SD = 2.83)	4.23 (SD = 2.14)	0.272; 0.206	26.47 (SD = 9.49)	30.03 (SD = 8.70)	−0.391; 0.083
Does not do sports	14 (5.3)	6.23 (SD = 3.47) *	4.72 (SD = 2.77)	0.674; 0.001	26.11 (SD = 9.14)	33.11 (SD = 10.25) *	−0.721; 0.050
Daily TV/Movie Viewing Time							
Max 1 h	30 (30.9)	5.14 (SD = 3.14)	4.91 (SD = 2.74)	0.078; 0.762	23.43 (SD = 8.11)	24.83 (SD = 8.61)	−0.167; 0.514
1–2 h	19 (19.6)	4.94 (SD = 2.77)	4.14 (SD = 2.24)	0.318; 0.334	26.75 (SD = 9.76)	29.15 (SD = 8.56)	−0.261; 0.425
More than 2 h	27 (27.8)	6.77 (SD = 4.11) *	4.12 (SD = 3.87)	0.664; 0.050	29.17 (SD = 10.74)	37.27 (SD = 9.90) *	−0.784; 0.005
I do not watch daily	21 (21.6)	5.03 (SD = 3.74)	4.13 (SD = 2.58)	0.280; 0.365	17.12 (SD = 9.07)	27.42 (SD = 8.74)	−0.599; 0.102
Internet Use Before Sleep							
Does not use the internet	13 (13.4)	3.65 (SD = 1.95)	2.93 (SD = 1.48)	0.416; 0.299	14.11 (SD = 6.13)	20.98 (SD = 7.87)	−0.974; 0.065
10–15 min	20 (20.6)	5.21 (SD = 3.23)	4.34 (SD = 2.71)	0.292; 0.361	22.49 (SD = 9.92)	29.99 (SD = 9.10)	−0.599; 0.076
More than 30 min	21 (21.6)	5.89 (SD = 3.93)	4.87 (SD = 2.43)	0.312; 0.317	22.57 (SD = 9.73)	29.91 (SD = 9.49)	−0.764; 0.071
More than 60 min	43 (44.3)	6.99 (SD = 4.11) *	5.02 (SD = 2.41)	0.603; 0.001	38.49 (SD = 9.79) *	31.94 (SD = 10.42)	0.648; 0.001
PC, Laptop, and Tablet Use							
Max 1 h	8 (8.3)	6.01 (SD = 4.10)	5.14 (SD = 3.78)	0.221; 0.665	22.41 (SD = 6.48)	28.21 (SD = 6.93)	−0.865; 0.105
1–2 h	24 (24.7)	4.07 (SD = 2.12)	3.41 (SD = 2.02)	0.319; 0.275	22.16 (SD = 8.02)	27.01 (SD = 8.25)	−0.596; 0.060
More than 2 h	65 (67.0)	6.24 (SD = 3.71) *	4.11 (SD = 2.51)	0.672; 0.002	35.12 (SD = 9.74) *	29.17 (SD = 10.72)	0.614; 0.001

* Statistically significant difference (*p* < 0.05).

**Table 2 healthcare-13-02272-t002:** Comparison of online vs. in-person education on sleep quality and perceived stress.

Dimension	Online Education (Remote Learning)	Frontal Education (In-Person Learning)
**Pros**	- Greater flexibility in time management - Elimination of commute - Potential for individualized pacing	- Structured schedule provides routine - Increased peer and faculty interaction - Easier access to academic support
**Cons**	- Higher AIS scores → poorer sleep quality (M = 5.47 vs. 4.25; *p* = 0.001) - Increased screen time, disrupted circadian rhythm - Reduced physical activity	- Higher PSS scores → increased stress (M = 29.48 vs. 24.31; *p* < 0.05) - Rigid scheduling - Greater academic pressure
**Impact on Sleep Quality**	Negative: Associated with insomnia, sleep dissatisfaction, and delayed bedtimes	Relatively better: Improved sleep quality compared to online period
**Impact on Perceived Stress**	Lower: Reduced psychological stress during remote learning phase	Higher: Greater perceived stress linked to academic workload and expectations
**Behavioral Risk Factors**	- Prolonged internet use before bed (e.g., >60 min) correlated with worse AIS and PSS scores	- Reduced exercise during busy semesters can worsen stress and sleep
**Academic Performance Influence**	- No significant effect on AIS or PSS scores	- Same: Grade average did not significantly moderate stress or sleep outcomes
**Mental Health Considerations**	- Blurred work–life boundaries may increase long-term mental strain despite lower acute stress	- Structured demands may foster burnout without adequate coping strategies
**Key Challenges**	- Maintaining sleep hygiene - Reducing screen time before bed	- Managing academic stress - Balancing institutional and clinical obligations

## Data Availability

The data presented in this study are available on request from the corresponding author due to privacy and ethical restrictions related to students’ confidentiality.

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
