# Peer review of "Comparing Insomnia and Perceived Stress in Online vs. Frontal Learning: Psychometric Evaluation in a Health Sciences Cohort"

_healthcare, 2025, doi:10.3390/healthcare13182272_

Round 1

Reviewer 1 Report

Comments and Suggestions for Authors

Review of the Manuscript:

Comparing Insomnia and Perceived Stress in Online vs. Frontal Learning: Psychometric Evaluation in a Health Sciences Cohort

In this manuscript, the authors evaluated the quality of sleep and the perceived stress level in health sciences students during online and traditional classes.

The general impression is that this manuscript is well-structured and easy to read; however, several critical issues need to be addressed before the manuscript can be considered for publication. The primary concerns are the short duration between the two measurements and the small sample size. These issues must be thoroughly addressed. But if we disregard that, I will have several suggestions for a major revision.

Namely, in the methodology, the authors must provide a clear and detailed description of the teaching modalities. It is essential to specify the duration and method of implementation for both the online and traditional classes. The distinction between synchronous and asynchronous online teaching is crucial, as each can have a different impact on sleep quality and stress levels. The manuscript mentions two distinct periods, but it is unclear how long the students were exposed to one type of teaching and how long to the other. This information is vital for contextualising the results.

Also, in methodology, the authors should state how the adequate sample size was determined. Furthermore, it is necessary to provide details regarding the total number of students at the University who met the criteria for inclusion in the study, and what percentage of them participated. How many students dropped out? Has anyone participated in the first phase, but not in the second? It should be stated precisely.

The current presentation of results is inefficient. A single figure and text are used for a portion of the results, while other critical findings are relegated to Supplement 1, which is inaccessible to me as a reviewer. There is no reason for the supplement. Namely, the use of a supplement for core results is not justified. The findings on differences in sleep quality and perceived stress levels across observed variables should be presented in a clear and concise table within the main body of the manuscript.

The discussion section refers to "Table 1," but this table is not present in the text. Please ensure that the reference table is appropriately integrated into the discussion.

The references are carefully selected and up-to-date, demonstrating a good grasp of the relevant literature.

I recommend a major revision with the expectation that the authors will comprehensively address the above-mentioned points.

Author Response

Dear Reviewer!

Thank you for your time and expertise.

Regarding your comments:

In this manuscript, the authors evaluated the quality of sleep and the perceived stress level in health sciences students during online and traditional classes.

The general impression is that this manuscript is well-structured and easy to read; however, several critical issues need to be addressed before the manuscript can be considered for publication. The primary concerns are the short duration between the two measurements and the small sample size. These issues must be thoroughly addressed. But if we disregard that, I will have several suggestions for a major revision.

Namely, in the methodology, the authors must provide a clear and detailed description of the teaching modalities. It is essential to specify the duration and method of implementation for both the online and traditional classes. The distinction between synchronous and asynchronous online teaching is crucial, as each can have a different impact on sleep quality and stress levels. The manuscript mentions two distinct periods, but it is unclear how long the students were exposed to one type of teaching and how long to the other. This information is vital for contextualising the results. - WE'VE ADDED INFORMATIONS AND CORRECTED THE MENTIONED PARTS.

Also, in methodology, the authors should state how the adequate sample size was determined. Furthermore, it is necessary to provide details regarding the total number of students at the University who met the criteria for inclusion in the study, and what percentage of them participated. How many students dropped out? Has anyone participated in the first phase, but not in the second? It should be stated precisely. - WE'VE TRIED TO IMPROVE THE DOCUMENT

The current presentation of results is inefficient. A single figure and text are used for a portion of the results, while other critical findings are relegated to Supplement 1, which is inaccessible to me as a reviewer. There is no reason for the supplement. Namely, the use of a supplement for core results is not justified. The findings on differences in sleep quality and perceived stress levels across observed variables should be presented in a clear and concise table within the main body of the manuscript. - THANK YOU, TABLE HAS BEEN ADDED

The discussion section refers to "Table 1," but this table is not present in the text. Please ensure that the reference table is appropriately integrated into the discussion. - THANK YOU, IT IS NOW CORRECTED

The references are carefully selected and up-to-date, demonstrating a good grasp of the relevant literature. - THANK YOU

I recommend a major revision with the expectation that the authors will comprehensively address the above-mentioned points.

Reviewer 2 Report

Comments and Suggestions for Authors

The paper discusses an important topic with good structure and content overall. Here are few suggestions that I believe should be addressed to improve the quality of the paper.

  1. In the introduction, please clarify how this follow-up study expands upon or differs from past research, particularly reference 6, which appears to be a previous study by the same authors. Also, elaborate more in lines 57 to 63 on the rationale for why these two modes may differentially affect stress and sleep and state the expected directional impact of each learning mode on stress and sleep to give the reader a better idea into the study's aims.
  2. in the methods section, provide the number of eligible students and response rate and state whether there were any unmatched or excluded cases due to incorrect code entry, and how these were handled (lines 162-168). Additionally, please include mor details on the development and validation of the lifestyle questionnaire. For example, explain whether items were piloted, and if internal consistency or item analysis was conducted. The statistical analysis seems ok, but I’d like to see the effect sizes (Cohen’s d) for primary comparisons of AIS and PSS between the two time points.
  3. the results section has strong information overall. When comparing AIS with PSS, please include the effect sizes for these differences to indicate practical significance (lines 189-195). In lines 198 to 216,the breakdown by gender, physical activity, and screen time is good, but it seems to me the interpretation is deferred to the discussion. So, please consider including a summary table of these subgroup comparisons and clarify whether these were pre-specified subgroups or exploratory analyses.
  4. Discussion: kindly discuss the potential contribution of irregular routines, unstructured schedules, and device use with more depth. In lines 238 to 2422, elaborate more on institutional demands (e.g., exams, clinical rotations) that may exacerbate stress. Moreover, in the limitation setion of the discussion, please acknowledge that self-reported measures may be subject to recall or desirability bias.
  5. In the conclusion, add a statement about practical implications for university health services, such as the promotion of sleep hygiene and digital wellness programs, and briefly emphasise that findings are specific to a single institution and health science student population, and may not generalize across disciplines.

Author Response

Dear Reviewer!

We would like to thank you for your time and expertise.

Regarding your comments:

The paper discusses an important topic with good structure and content overall. Here are few suggestions that I believe should be addressed to improve the quality of the paper.

  1. In the introduction, please clarify how this follow-up study expands upon or differs from past research, particularly reference 6, which appears to be a previous study by the same authors. Also, elaborate more in lines 57 to 63 on the rationale for why these two modes may differentially affect stress and sleep and state the expected directional impact of each learning mode on stress and sleep to give the reader a better idea into the study's aims. - WE'VE ADDED SOME RELEVANT INFORMATION ABOUT THE MENTIONED PARAGRAPH
  2. in the methods section, provide the number of eligible students and response rate and state whether there were any unmatched or excluded cases due to incorrect code entry, and how these were handled (lines 162-168). Additionally, please include mor details on the development and validation of the lifestyle questionnaire. For example, explain whether items were piloted, and if internal consistency or item analysis was conducted. The statistical analysis seems ok, but I’d like to see the effect sizes (Cohen’s d) for primary comparisons of AIS and PSS between the two time points. - WE'VE TRIED TO EXPLAIN OUR RESEARCH'S DETAILS AS YOU SUGGESTED.
  3. the results section has strong information overall. When comparing AIS with PSS, please include the effect sizes for these differences to indicate practical significance (lines 189-195). In lines 198 to 216,the breakdown by gender, physical activity, and screen time is good, but it seems to me the interpretation is deferred to the discussion. So, please consider including a summary table of these subgroup comparisons and clarify whether these were pre-specified subgroups or exploratory analyses. - WE'VE ADDED THE TABLE OF RESULTS.
  4. Discussion: kindly discuss the potential contribution of irregular routines, unstructured schedules, and device use with more depth. In lines 238 to 2422, elaborate more on institutional demands (e.g., exams, clinical rotations) that may exacerbate stress. Moreover, in the limitation setion of the discussion, please acknowledge that self-reported measures may be subject to recall or desirability bias. - THANK YOU, WE'VE ADDED MORE EXPLANATIONS.
  5. In the conclusion, add a statement about practical implications for university health services, such as the promotion of sleep hygiene and digital wellness programs, and briefly emphasise that findings are specific to a single institution and health science student population, and may not generalize across disciplines. - THANK YOU, WE'VE TRIED OUR BEST AND REWROTE THE MENTIONED SECTION

Round 2

Reviewer 1 Report

Comments and Suggestions for Authors

Second review for manuscript titled Comparing Insomnia and Perceived Stress in Online vs. Frontal Learning: Psychometric Evaluation in a Health Sciences Cohort

After thoroughly reading the revised manuscript, I conclude that the authors made every effort to incorporate the suggested changes into the manuscript, aiming to enhance its scientific acceptability and accuracy. And that is commendable; however, in my opinion, this version of the manuscript requires minor revision before acceptance.

Namely, the methodology remains unclear regarding the teaching format for students during March, specifically whether they had online instruction from February 1 to March 1 and traditional instruction from April 1 to May 26. Furthermore, the authors state, "Students were therefore exposed to the online teaching modality for approximately 4 weeks before transitioning to traditional teaching". Does this mean that students had online classes until April 1? Not clearly explained. Please clarify the timeline explicitly.

The sentence from lines 116 and 117 should be deleted, because it appears to be duplicated.

In the results, Table 1, which was added in response to prior feedback, needs to be supplemented with essential statistical data. Namely, it is necessary to insert columns for p-value and Cohen's d value. Without that data in the table, the text in the results remains difficult to follow.

And finally, Table 2 in the discussion is not well-connected to the text. Namely, the table does not represent what the authors refer to as "Taken together, these findings underscore the need for context-sensitive mental health strategies that address the specific demands of each learning environment. (Table 2). I suggest that the discussion should include a sentence that points to the facts listed in the table.

Author Response

Dear Reviewer!

Thank you for your expertise - we tried to change our article regarding your comments:

After thoroughly reading the revised manuscript, I conclude that the authors made every effort to incorporate the suggested changes into the manuscript, aiming to enhance its scientific acceptability and accuracy. And that is commendable; however, in my opinion, this version of the manuscript requires minor revision before acceptance. - THANK YOU

Namely, the methodology remains unclear regarding the teaching format for students during March, specifically whether they had online instruction from February 1 to March 1 and traditional instruction from April 1 to May 26. Furthermore, the authors state, "Students were therefore exposed to the online teaching modality for approximately 4 weeks before transitioning to traditional teaching". Does this mean that students had online classes until April 1? Not clearly explained. Please clarify the timeline explicitly. - THE ONLINE TEACHING WAS FROM 1ST OF FEBRUARY TO 1ST OF MARCH, THAN STUDENTS CAME BACK TO "NORMAL" LECTURING. WE'VE COLLECTED DATA FROM 1ST OF APRIL TO 26TH OF MAY (THE END OF THE SEMESTER, BEFORE EXAM PERIOD).

The sentence from lines 116 and 117 should be deleted, because it appears to be duplicated. - I CANNOT SEE, I'M TRULY SORRY FOR THAT.

In the results, Table 1, which was added in response to prior feedback, needs to be supplemented with essential statistical data. Namely, it is necessary to insert columns for p-value and Cohen's d value. Without that data in the table, the text in the results remains difficult to follow. - ALL ADDED

And finally, Table 2 in the discussion is not well-connected to the text. Namely, the table does not represent what the authors refer to as "Taken together, these findings underscore the need for context-sensitive mental health strategies that address the specific demands of each learning environment. (Table 2). I suggest that the discussion should include a sentence that points to the facts listed in the table. - WE'VE PUT A SENTENCE REGARDING YOUR COMMENT

Reviewer 2 Report

Comments and Suggestions for Authors

My comments have been addresses. thank you. 

Author Response

Dear Reviewer!

Thank you so much for your tuly positive feedback.